# Extracellular Matrix Components Regulate Bone Sialoprotein Expression in MDA-MB-231 Breast Cancer Cells

**DOI:** 10.3390/cells10061304

**Published:** 2021-05-24

**Authors:** Florian Keller, Roman Bruch, Franziska Clauder, Mathias Hafner, Rüdiger Rudolf

**Affiliations:** 1Institute of Molecular and Cell Biology, Mannheim University of Applied Sciences, 68163 Mannheim, Germany; f.keller@hs-mannheim.de (F.K.); r.bruch@hs-mannheim.de (R.B.); m.hafner@hs-mannheim.de (M.H.); 2Institute of Medical Technology, Heidelberg University and Mannheim University of Applied Sciences, 68167 Mannheim, Germany; 3Immundiagnostik AG, 64625 Bensheim, Germany; franziska.clauder@immundiagnostik.com

**Keywords:** basal membrane extract, bone sialoprotein, breast cancer, extracellular matrix, MDA-MB-231, proteolysis, spheroid

## Abstract

Bone sialoprotein (BSP) has become a target in breast cancer research as it is associated with tumor progression and metastasis. The mechanisms underlying the regulation of BSP expression have been largely elusive. Given that BSP is involved in the homing of cancer cells in bone metastatic niches, we addressed regulatory effects of proteolytic cleavage and extracellular matrix components on BSP expression and distribution in cell culture models. Therefore, MDA-MB-231 human breast cancer cells were kept in 2D and 3D spheroid cultures and exposed to basement membrane extract in the presence or absence of matrix metalloproteinase 9 or the non-polar protease, dispase. Confocal imaging of immunofluorescence samples stained with different antibodies against human BSP demonstrated a strong inducing effect of basement membrane extract on anti-BSP immunofluorescence. Similarly, protease incubation led to acute upregulation of anti-BSP immunofluorescence signals, which was blocked by cycloheximide, suggesting de novo formation of BSP. In summary, our data show that extracellular matrix components play an important function in regulating BSP expression and hint at mechanisms for the formation of bone-associated metastasis in breast cancer that might involve local control of BSP levels by extracellular matrix degradation and release of growth factors.

## 1. Introduction

Breast cancer is one of the most lethal illnesses for women and the most common type of malignancy besides skin cancer. Only 26% of women with distant malignancies survive a 5-year time-period after diagnosis [1]. Furthermore, 65–75% of metastases spread into bone tissue [2] and develop features that lead to either osteolytic osteoporosis, osteoblastic sclerosis, or both. In osteolytic regions, niche formation for metastatic apposition is a common mechanism [3]. According to current knowledge, bone resorption is not mediated by the cancer cells themselves, but primarily involves osteoclast activation [4]. Mechanistically, some evidence suggests a role of growth factors such as transforming growth factor beta (TGFβ), vascular endothelial growth factor (VEGF) and interleukins, such as interleukin-6 (IL-6), in the underlying remodeling processes at osteolytic metastatic sites [5]. However, the origin of these factors and the rationale for their release are still largely elusive. Investigations in nude mice indicate that homing and niche formation of breast cancer cells are independent of estrogen receptor status [6] and thus, the mode of action might be transferable to different breast cancer types. Regularly, breast cancer lesions are located within trabecular bone regions that are rich in osteoblasts and micro-vessels and may overlap with the hematopoietic stem cell niche [7]. Although trabecular bone has the same lamellar structure as compact bone [8], slight differences in remodeling processes of these two bone types may give a hint why trabecular bone is preferred for metastasis. Indeed, while general features of bone extracellular matrix (ECM) are similar, such as the presence of collagenous (type 1, type 3 and type 5 collagen) and non-collagenous proteins (proteoglycans, glycoproteins and glyco-phosphoprotein) as well as inorganic ECM (hydroxyapatite) [9], trabecular bone shows, in addition, increased levels of Wnt3a, β-catenin, osteocalcin and RANK-L [10]. Consequently, interference with these pathways was found to have a huge impact on both, osteoblastic bone formation and osteoclastic bone resorption. For example, inhibition of β-catenin-mediated canonical Wnt-signaling [11] led to decreased osteoblast differentiation and induced resorption of bone mass [12]. In addition, RUNX2-mediated signaling cascades regulated expression of small integrin-binding ligand, N-linked glycoproteins (SIBLING) in early-stage bone formation processes [13], making them a crucial component of the ECM.

While bone-related cancer treatments often start at late stages by, e.g., interfering with RANK-L signaling, the disturbance of bone homeostasis that leads to niche formation presumably starts early in the tumor development [14]. Physiologically, bone remodeling is a 5-stage process, consisting of quiescent, activation, resorption, formation, and mineralization phases [15]. Osteoclasts are key players in the bone resorption phase, during which small lacunae are formed through the removal of pre-existing mineral and organic components like hydroxyapatite and collagens. To that end, osteoclasts express various matrix metalloproteases (MMPs) that do not only participate in initial lacuna formation, but also serve additional functions such as cell recruitment (MMP-9), survival (MMP-9), ECM degradation (MMP-12), macrophage fusion and adhesion (MMP-14) or maturation (MMP-7 and MMP-14 through RANK/RANK-L guidance) [16]. Thus, MMPs are capable of interfering with various ECM components. Especially, osteopontin and bone sialoprotein (BSP), members of the SIBLING glycoprotein family, are known to have multiple MMP-related binding partners that might interfere with bone tissue homeostasis to support metastatic niche generation [17]. Both proteins are markers for osteoblast differentiation [18] and they are crucial for bone-tissue remodeling under physiological and tumor-associated conditions. Notably, tumor cells and their microenvironment synergistically interact with each other during cancer progression and this involves positive feedback-loops that can trigger severe effects based on minimal alterations of cancer and stroma cell interactions [19]. Unfortunately, in currently available cell-culture models, these small changes might be masked by undefined supplements, such as Matrigel or basal membrane extract (BME). These biomaterials contain not only structurally important proteins, such as collagens, but also largely undefined mixes of signaling molecules with significant batch-to-batch variances [20]. To address potential mutual interactions between tumor cells and their microenvironment, this work focused on effects of extracellular matrix components and proteases on cellular responses, with a focus on the SIBLING protein, BSP.

## 2. Materials and Methods

### 2.1. Cell Culture

To cultivate the MDA-MB-231 human breast cancer cell line, Dulbecco’s modified Eagle medium (Capricorn Scientific, Ebsdorfergrund, Germany; DMEM-HPA; Lot CP18-2096) was supplemented with 10% fetal bovine serum (Capricorn; FBS-12B; Lot CP16-1422), 1% penicillin/streptomycin (Sigma-Aldrich, St. Louis, MO, USA; P4333-100ML; Lot #91675) and 1% minimum essential medium nonessential amino acids (Sigma-Aldrich; M7145-100ML; Lot RNBJ0616). Passaging was done twice a week with seeding densities of 1 × 10^6^ cells/T75 flask.

For two-dimensional cell analysis, circular cover slips (12 mm; VWR; ECN 631-1577; Lot #43395 819) were sterilized and placed in culture dishes (Greiner Bio-One, Frickenhausen, Germany; 100 × 20 PS; 664 160) prior to seeding. Then, 1.2 × 10^6^ cells were cultured in 10 mL of media with appropriate supplements for 4 d prior to fixation.

3D cell culture was based on spheroids grown in 96-well cell-repellent microplates (Greiner; PS U-bottom; 650970; Lot E20063QS). Therefore, 4000–8000 breast cancer cells were set into one well with a total volume of 150 µL and supplemented with 2.5% BME (Bio-Techne, Minneapolis, MN, USA; Cultrex; BME Pathclear; 3432-005-01; Lot 1514944) or 5 mM type 1 collagen (Roche Diagnostics, Basel, Switzerland; 11179179001; Lot #38429220) as required. To ensure a complete integration of all cells into the spheroids, cell aggregation was supported by initial centrifugation for 6 min at 500 rcf.

Dispase (EMD Millipore, Burlington, MA, USA; SCM133; Lot #3168044) treatment studies were accomplished after 4 d of culturing. Culture controls were fixed directly prior to treatment. For mode of action studies, cultures were washed twice with phosphate buffered saline (PBS), then exposed to dispase for different time periods (2–10 min), rinsed with FCS-containing media to stop enzymatic induction, washed again twice and then either fixed or contingently further cultivated in media that ranged from common culture media to secondary treatment supplementing with 5 µM cycloheximide (Sigma-Aldrich; Cycloheximide Solution; 18079-10X10ML-F; Lot #BCCB2943). The elapsed time from when protease treatment stopped to fixation was about 3 min. Additionally, treatment controls were handled in parallel to the secondary treatment studies with common media exchanges and no added drugs.

Furthermore, 400 ng/mL MMP-9 (Abcam, Cambridge, MA, USA; ab168863; LotGR3381311-1) treatment, stopping, washing and further processing—including timelines—were performed as for dispase exposure. The culture was kept in MMP-9-containing FCS-free media for appropriate time periods (2–12 min), then rinsed with FCS-containing media, washed twice with PBS, and further cultivated for a total cultivation time of 25 min after first MMP-9 contact. Then all samples were fixed at once.

### 2.2. Sample Preparation

For immunostaining analysis, samples were washed in PBS, fixed with 4% paraformaldehyde in PBS (PFA) and washed again prior to further handling.

2D samples are permeabilized (0.4% Triton X-100 in deionized water/5 min) and blocked with bovine serum albumin fraction V (BSA; Carl Roth, Karlsruhe, Germany; 8076.3; Lot 479289436) buffer (3% in PBS/1 h) at room temperature (RT). Primary antibodies were diluted in BSA buffer and applied overnight (ON) at 4 °C, while secondary antibodies were incubated at RT (2 h). For intermediate washing, PBS was used (3 × 5 min/RT). Finally, samples were mounted using 10% Mowiol (Sigma-Aldrich).

Whole-mount spheroid samples were stained and cleared with glycerol as published earlier [21,22]. In brief, fixed samples were permeabilized with 2% Triton X-100 (2 min/RT) following an incubation with penetration buffer (10% dimethyl sulfoxide (DMSO), 0.3 M glycine and 0.2% Triton X-100 in PBS/2 h). Blocking buffer (10% DMSO, 1% BSA and 0.2% Triton X-100 in PBS/2 h) was applied prior to staining through appropriate immunoglobulin incubations in antibody buffer (5% DMSO, 1% BSA, 0.2% Tween 20 and 10 µg/mL heparin in PBS/ON). Washing buffer (0.2% Tween 20 and 10 µg/mL heparin in PBS) was applied three times after each antibody incubation step. For terminal clearing, samples were incubated with 88% glycerol in deionized water with the refractive index of 1.458 for 24 h. All buffer and clearing incubations were performed on a roller mixer at 37 °C. For storage, stained samples were kept in the dark at 4 °C.

### 2.3. Antibodies and Dyes

Primary antibodies used had the following dilutions and specifications: AF165 (1:250; Immundiagnostik; AF165 (a-BSP); human monoclonal; Lot RP-SZ_423/03), FP21 (1:250; Immundiagnostik; FP-21 (a-BSP); mouse monoclonal; Lot AK679/01B.2), IDK1 (1:100; Immundiagnostik; IDK-1 (a-BSP); rat monoclonal; Lot AK606/05B.4), GM130 (1:400; BD Bioscience; 610822; mouse monoclonal; Lot #262), IGF1 (1:250; Invitrogen; PA5-27207; rabbit polyclonal; Lot TK2672961D), MMP-9 (1:250; abcam; ab38898; rabbit polyclonal to MMP9; Lot GR3204084-22), RUNX2 (1:500; Cell Signaling; D1L7F; rabbit monoclonal; Lot #2), and TGFβ (1:500; abcam; ab92486; rabbit polyclonal to TGF beta 1; Lot GR3237963-2).

Secondary antibodies were incubated together with nuclear dye as follows: Anti-Rat 488 (1:500; Invitrogen; Alexa Fluor 488 donkey anti-rat; A21208; Lot #1810471), Anti-Mouse 555 (1:1000; Invitrogen; Alexa Fluor 555 goat anti-mouse IgG (H + L); A21424; Lot #2123594), Anti-Mouse 647 (1:1000; Invitrogen; Alexa Fluor 647 donkey anti-mouse IgG (H + L); A31571; Lot #2045337), Anti-Human 488 (1:1000; Invitrogen; Alexa Fluor 488 goat anti-human IgG (H + L); A11013; Lot #2110842), Anti-Human 647 (1:1000; Invitrogen; Alexa Fluor 647 goat anti-human IgG (H + L); A21445; Lot #2160390), Anti-rabbit 647 (1:1000; Invitrogen; Alexa Fluor 647 goat anti-rabbit IgG (H + L); A21246; Lot #55002A), and DAPI (1:1000; Roche Diagnostics; 10236276001; Lot #28114320).

### 2.4. Data Acquisition and Analysis

Before starting the imaging, samples were stored within the same room to match their temperatures. Confocal scans (Leica Microsystems, Mannheim, Germany; SP8; objectives HC PL APO 20 x/0.75 IMM CORR and HC PL APO CS2 63 x/1.2 W CORR; software Leica LAS-X 3.3.0) were captured with a resolution of 1024 × 1024 pixels when using 20× and 2048 × 2048 pixels for 63× objective pictures. Z-step size was set to 0.4 µm for 2D analysis and 1 µm in the case of whole mount spheroid investigations. To reduce the impact of glass surface related background, only signals above the nuclei center were included for quantifications. Therefore, we automatically segmented the DAPI signal and averaged their morphological centers as stack borders to remove all lower planes. Data analysis was performed using ImageJ software (National Institute of Health, Wisconsin, WI, USA; v1.52) for either summed up or mean signal intensity measurements within cell culture contained regions of interest within confocal scans.

For statistics, GraphPad Prism 7 software (GraphPad, San Diego, CA, USA) was used. Normal distribution was ensured with the Kolmogorov–Smirnov test. One-way ANOVA with Holm–Sidak multiple comparison was applied. For the analysis, an appropriate number of experiments were executed (see corresponding figure legend) and at least 10 random cells were analyzed per experiment. The mean of means ensured a Gaussian distribution in this case and multiple comparison *t*-tests without assumptions were applied. Significance was defined based on *p*-values (* *p* < 0.05, ** *p* < 0.01, *** *p* < 0.001, **** *p* < 0.0001).

## 3. Results

### 3.1. BME and Collagen 1 Enhance BSP Expression in MDA-MB-231 Spheroids

Earlier investigations suggested a high impact of media composition on growth and morphology of MDA-MB-231 spheroids [23]. Since we wanted to gauge the role of ECM components on breast cancer cell growth and BSP expression, 3D culture conditions with different extracellular supplements were compared. Therefore, 8000 MDA-MB-231 cells were seeded on ultra-low attachment plates and supplemented with either 2.5% BME, 5 µg/mL collagen type 1 or nothing. Spheroid growth and BSP expression were analyzed after four days of culturing, because at later time points, cultures in the absence of either collagen or BME started to disintegrate. Thus, whole-mount confocal imaging of fixed and permeabilized spheroid samples was performed upon staining with DAPI and two different anti-BSP antibodies, AF165 and FP21. To visualize the entire upper half of each spheroid, samples were optically cleared. For quantitative assessment, the sum of immunofluorescence signal per spheroid was normalized to the corresponding DAPI fluorescence. Qualitative analysis revealed that the subcellular localization of BSP immunofluorescence signals differed between AF165 and FP21 staining (Figure 1A). Specifically, they appeared to label cell boundaries and perinuclear regions, respectively. Yet, despite the differential subcellular localization, the general trend of signal intensities was similar between the two antibody staining groups (Figure 1B). Indeed, for both markers, BME supplementation led to the highest signal increase compared to the condition without supplement. Collagen 1 resulted in an intermediate effect: compared to non-supplemented cultures, there were slightly rising levels of FP21 signals and unaltered AF165 signals, but signal intensities were lower than with BME.

### 3.2. Both BME and Short-Term Protease Treatment Enhance BSP Immunofluorescence

To investigate the effects of BME on BSP expression in MDA-MB-231 spheroids in more detail, further experiments in the absence and presence of BME were performed, but now in 2D adherent cultures. As for spheroids, adherent cultures of MDA-MB-231 cells also showed significantly higher fluorescence intensity signals for both, AF165 and FP21, in the presence of BME (+2.5% BME, Figure 2) than in their absence (culture control, Figure 2). Next, since it is known that local proteolytic activities occur during metastatic niche formation, we also assessed whether these could be mimicked in vitro and might affect BSP expression. Therefore, dispase treatments were set up to digest the ECM of the 2D culture without inducing cell detachment. Thus, after four days in 2D culture—in the absence of BME—MDA-MB-231 cells were exposed to dispase for up to 6 min. This time range was chosen, because longer incubation resulted in a complete cell loss (data not shown). Samples were then fixed, stained with anti-BSP-antibodies, AF165 and FP21 (Figure 2A), and mean signal intensities of confocal scans were quantitatively analyzed. On average, the time from stopping dispase treatment to fixation of the cells lasted about 3 min. As shown in Figure 2B,C, this revealed increased immunofluorescence signal intensities at the cell level for both antibodies upon dispase treatment. The amount of induction correlated with dispase exposure times and was independent of operative stress since treatment controls with parallel washing and media lacking dispase did not result in enhanced immunofluorescence signals.

### 3.3. Dispase Appears to Induce BSP Biosynthesis in MDA-MB-231 Cells

To understand, whether the enhanced BSP immunofluorescence signals in the presence of dispase were due to increased biosynthesis, BSP exocytosis, or both, additional tests were performed. Specifically, to test the effect of dispase on BSP secretion, MDA-MB-231 2D cultures were treated with or without dispase for 6 min, incubated with normal medium or cycloheximide for 0 or 4 h and then fixed. Subsequently, samples were immunostained in the absence of permeabilization with the two anti-BSP antibodies, AF165 and IDK1 (Figure 3A). In this and the following experiment, IDK1 anti-BSP antibody was used instead of FP21, because FP21 showed an intracellular fluorescence signal (Appendix A) and this was incompatible with the analysis of BSP release of non-permeabilized cells. Conversely, both, AF165 and IDK1, yielded extracellular signals and were, therefore, both employed here to consolidate the findings. Representative images of this test are shown in Figure 3A. Quantitative analysis of the data revealed for AF165, that dispase treatment resulted in a significant decrease of signal intensity per cell when comparing the culture control with treated conditions (Figure 3B). The same trends were observed for IDK1, but no significances could be found with this antibody (Figure 3C). A schematic of the experimental plans is depicted in Figure 3D.

Furthermore, to address the effect of dispase on BSP expression, the same experiments as in Figure 3 were carried out, but cells were now permeabilized upon fixation to also reveal intracellular BSP species. As illustrated with representative images in Figure 4A, confocal image analysis showed that dispase exposure for 6 min resulted in enhanced AF165 signal intensity per cell and this further increased upon 4 h of post-incubation in media (Figure 4B, compare bars 1, 2 and 3). Application of cycloheximide blocked any such signal increase (Figure 4B, bar 5). Similar trends resulted from IDK1-signal quantification, although all differences were less pronounced (Figure 4C). A schematic of the experimental plans is depicted in Figure 4D.

### 3.4. Regulatory Markers Are Consistent with Dispase-Induced BSP Expression

To corroborate de novo BSP protein expression upon dispase treatment, markers connected to BSP expression pathways were also examined. Thus, after dispase treatment for 6 min, cells were fixed, permeabilized, and stained for nuclei (DAPI), BSP (AF165, FP21) and, in addition, for IGF1, RUNX2, TGFβ or MMP-9. As illustrated in Figure 5, TGFβ and MMP-9 revealed a signal increase, similar to signals of AF165 and FP21. Furthermore, while RUNX2 showed a similar rising trend upon dispase treatment, IGF1 remained unchanged under these conditions.

### 3.5. Dispase Leads to an Increase of BSP Levels in MDA-MB-231 Spheroids

Next, both BME and dispase were applied to the MDA-MB-231 spheroid model. Furthermore, 4000 MDA-MB-231 cells were seeded into ultra-low attachment plates and supplemented with 2.5% BME to form spheroids. After 4 d in 3D culture, spheroids were exposed to dispase for up to 10 min. Then, samples were fixed, stained with AF165 and FP21 antibodies, and with DAPI. After optical tissue clearing, half of each spheroid was scanned by confocal imaging (Figure 6A). For quantification, the sums of immunofluorescence signals were normalized to that of DAPI. For AF165, this revealed an increase in signal intensity after 2 min and 4 min of dispase exposure, while longer incubation times returned signals to levels comparable to non-treated controls (Figure 6B). For FP21, similar trends were observed, but only the differences between signal intensities after 2 min and 4 min dispase treatment with 10-min exposed samples were significant (Figure 6C). As a further control, the sum of DAPI signal per spheroid was quantified to prove the integrity of the spheroids throughout investigation. This revealed no significant changes over the course of dispase treatment (Figure 6D). However, qualitatively, spheroid integrity and roundness started to decrease after 10 min of dispase. A schematic of the experimental plan is depicted in Figure 6E.

### 3.6. MMP-9 Exposure Increases BSP Signals in Adherent MDA-MB-231 Cultures

Finally, to substantiate the finding of a proteolysis-induced regulation of BSP levels with a physiologically relevant protease, we also tested the effect of MMP-9 incubation on anti-BSP fluorescence signal levels. Therefore, MDA-MB-231 cells were cultivated as adherent cultures for 4 d and then incubated with active human MMP-9 protein for up to 12 min. After washing, fixation and permeabilization, samples were stained for BSP (AF165 and FP21) as well as for MMP-9 and with DAPI. As shown by representative images (Figure 7A) and quantification of total intensity per cell (Figure 7B), anti-BSP fluorescence signals increased significantly after 2 min and 6 min of MMP-9 treatment for AF165 and FP21, respectively. Similarly, anti-MMP-9 fluorescence signals also augmented after 6 and 12 min of incubation.

## 4. Discussion

Cancer cell homing and niche formation are critical steps in metastasis. Enhanced levels of BSP correlate with the development of bone lesions [24]. To better understand the interaction between BSP and metastatic niche formation in bone, we addressed the effects of ECM components and extracellular proteolysis on BSP expression in breast cancer cells.

The present study showed that supplementation of MDA-MB-231 seeds with collagen 1 enhanced spheroid formation and BSP expression compared to similar cultures without any addition of ECM components. This suggests that structural support by a collagen mesh might affect cell-cell interaction and protein expression patterns. These effects were excelled by BME. Although BME, being a biological supplement, is known to show considerable batch to batch variability [25], it contains the major structural ECM components, collagen and laminin [26], as well as a less defined mix of growth factors. These include fibroblast growth factor (FGF), transforming growth factor beta (TGFβ), insulin-like growth factor (IGF) and epithelial growth factor (EGF) [27], all known to strongly affect cellular behavior and to interfere with BSP expression [28]. However, effects of adding either TGFβ or IGF-1 to the culture media in the absence of BME were inconsistent regarding BSP expression (not shown).

Next, given that either BME, Matrigel, or both, also contain proteolytic activity [29], and considering that cancer-cell homing is characterized by proteolytic preparation of the niche ECM [30], we hypothesized that these processes might directly or indirectly also affect the expression pattern of cancer-cell derived BSP. Fittingly, acute incubation with the proteases, dispase as well as MMP-9, the latter being a major protease expressed by metastatic cancer cells [31], led to an enhanced immunofluorescence staining signal upon use of different anti-BSP antibodies. The different anti-BSP antibodies employed in this study were chosen to validate each other by parallel use on the same samples. This was possible due to their different host origins: human, mouse and rat for AF165, FP21 and IDK1, respectively. Since AF165 staining appeared primarily at the outside of the cells and the rim of spheroids, thus, exposed to extracellular receptors (Appendix A), we hypothesize that the BSP species detected by this antibody participated in intercellular signaling. The same might be true for IDK1, because its staining was also localized to the membrane and its clinical relevance was suggested with MDA-MB-231 xenografts in nude rats; treatment with IDK1 led to decreasing tumor volumes [32]. In contrast, FP21 signals were primarily located in the Golgi apparatus and they were found in most cells throughout spheroids (Appendix A). This might indicate the ubiquitous and continuous presence of BSP-mRNA in MDA-MB-231 cells. One may speculate that BSP species recognized by FP21 were early forms undergoing processing in the Golgi apparatus, while those labeled by AF165 and IDK1 were more mature forms and secreted. However, further proof would be needed to confirm such an assumption. Currently, we can only speculate, whether and to what extent these data correlate to the observed effects of BME on the increase of anti-BSP immunofluorescence signals. Furthermore, it remains elusive, which proteolytic activity on BME or in a cancer niche might be relevant here. Yet, for MMP-9, it is known that it is poorly expressed in intact tissue but strongly upregulated upon injury [33] and relevant in bone resorption processes [34] through its role in osteoclast differentiation [35]. Increased levels of MMP-9 could simulate wounds and therefore stimulate various associated actions [36], such as activating substrates like VEGF [37] or TGFβ [38]. These, in turn, could further increase the amount of MMPs within direct cell proximity through triggering their expression [39] and might be relevant for regulating BSP expression.

Notably, the overt increase of BSP immunofluorescence signals upon acute dispase treatment was blocked by cycloheximide. Given that cycloheximide inhibits translational processes in general [40], this result suggests that proteolytic activities in the immediate vicinity of breast cancer cells might trigger feedback mechanisms that favor their homing and niche formation through a signal-transduction cascade that increases either acute protein expression, release of BSP, or in combination. This was corroborated by the basal availability of FP21 labeling throughout the spheroid, as this suggested constant BSP expression; thus, assuming the presence of BSP mRNA in the cell and an average translation elongation speed of 4–6 amino acids per second [41], de novo formation of BSP protein should be possible within roughly 1 min. This could be compliant with the time line of upregulation of anti-BSP immunofluorescence upon dispase and MMP-9 after only a few minutes, as observed in the present study. In reality, a few more minutes might have elapsed under these conditions, since protease inactivation, washing and fixation steps might have summed up to an additional 2–3 min. Furthermore, it cannot be excluded, that the protease treatment also led to a partially enhanced accessibility of the anti-BSP antibodies to their epitopes, at least for the extracellular signals of AF165 antibody.

Assuming a proteolytic activation of BSP expression, this might be linked to protease activated receptors (PAR). These can be cleaved by trypsin, but also by ECM-containing enzymes like proteinase 3 or MMP-9. They may activate various signaling pathways such as ion channels, ERK or Ras via G-Protein signal transduction and therefore have also been studied as targets for tumor treatment [42]. MDA-MB-231 cells express the PAR2-receptor, and its activity has been linked to fast cellular responses. Upon PAR-2 activation, induced migration of MDA-MB-231 cells was shown to start within 2 min, and within 5 min, trypsin-mediated protein synthesis based on pERK induction was observed [43]. Moreover, further patho-physiological effects have been linked to proteolytic activity. For example, in human bronchial epithelial cells, EGFR and TGF signaling were induced by protease exposure [44] and MMP-9 proenzyme induced angiogenesis via FGF activation [45]. Furthermore, MMP-9 supported pre-metastatic niche development upon induced overexpression in tumor stroma [46] and, thus, can be related to ECM-mediated stimuli. Therefore, IGF1 [47], RUNX2 and TGFβ [48], as well as MMP-9 were studied in this context. The fast upregulation of BSP observed in the present study is consistent with proteolytic activation of latent pro-enzymes of TGFβ [49] and MMP-9 [50], or with PAR-mediated signal transduction. Thus, this study points at how quickly local availability and activity of associated factors might change upon ECM induction.

## 5. Conclusions

In summary, this study reports that BSP protein expression in MDA-MB-231 breast cancer cells is enhanced upon incubation with collagen 1, basement membrane extract, and short-term proteolytic treatment using dispase or MMP-9. These findings are consistent with a mutual interaction between cancer cell and lesion site that could favor feedforward and feedback mechanisms during cancer cell homing, niche formation, or both.

## Figures and Tables

**Figure 1 cells-10-01304-f001:**
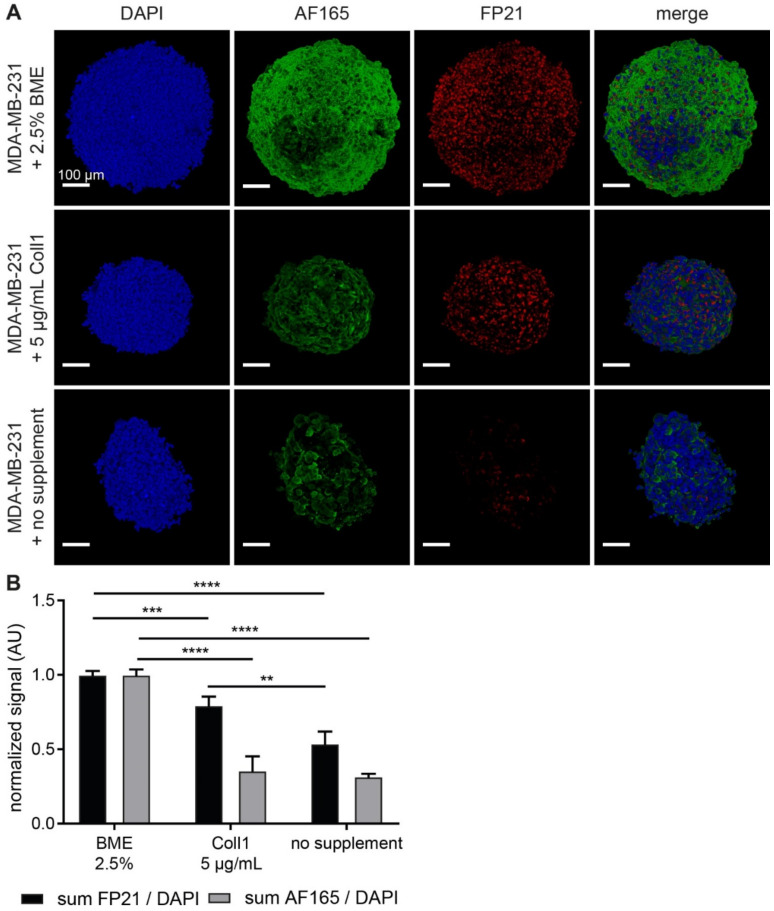
Basal membrane extract (BME) and collagen 1 enhance anti-BSP immunofluorescence signals in MDA-MB-231 spheroids. 8000 MDA-MB-231 cells were seeded in mono-culture spheroids and supplemented with 2.5% BME, 5 μg/mL collagen 1, or no additive upon seeding. After 4 d, spheroids were fixed and stained for DAPI and BSP (AF165 and FP21) as indicated. (**A**) Representative confocal image stacks of individual spheroids depicted as volume projections. (**B**) Graphs showing quantitative analysis of fluorescence intensities normalized with DAPI. Mean + SD (*n* = 4; ** *p* < 0.01, *** *p* < 0.001, **** *p* < 0.0001).

**Figure 2 cells-10-01304-f002:**
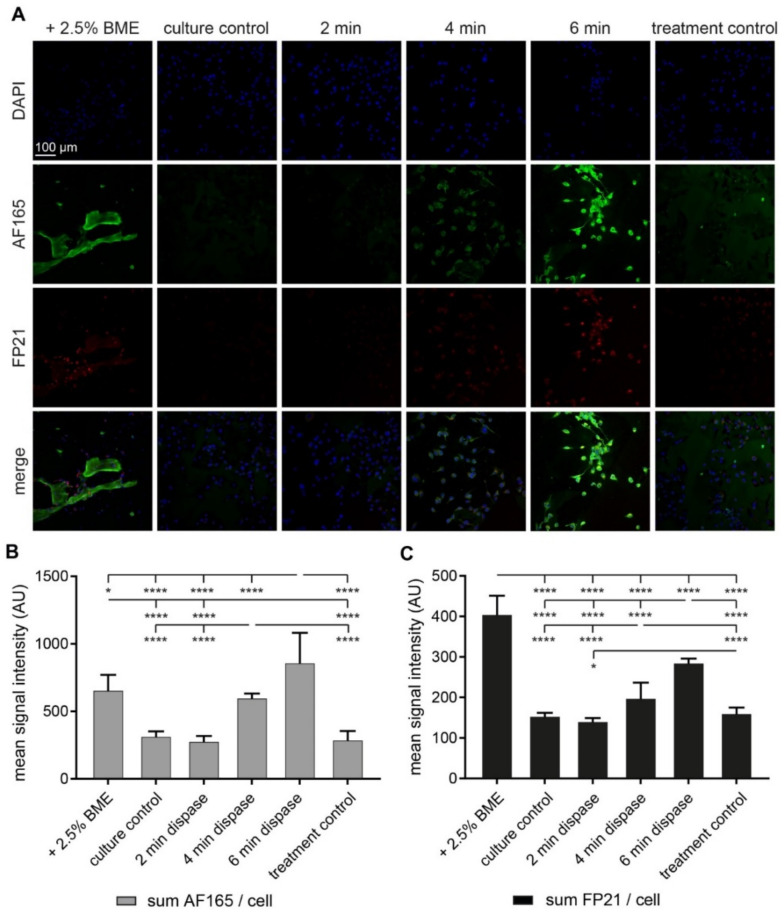
BME and short-term protease treatment enhance bone sialoprotein (BSP) immunofluorescence. 1,200,000 MDA-MB-231 cells were seeded in Petri dishes containing glass cover slips. After 2 d, one culture was supplemented with 2.5% BME within the media. Alternatively, no BME was added and after 4 d, cultures were treated with either dispase for exposure times as indicated, or with fresh media as treatment control. All cover slips were then fixed and stained for DAPI and human BSP (AF165 and FP21). (**A**) Representative confocal images in sum projections. (**B**,**C**) Graphs showing quantitative analysis of fluorescence intensities per cell. Mean + SD (*n* = 3; * *p* < 0.05, **** *p* < 0.0001).

**Figure 3 cells-10-01304-f003:**
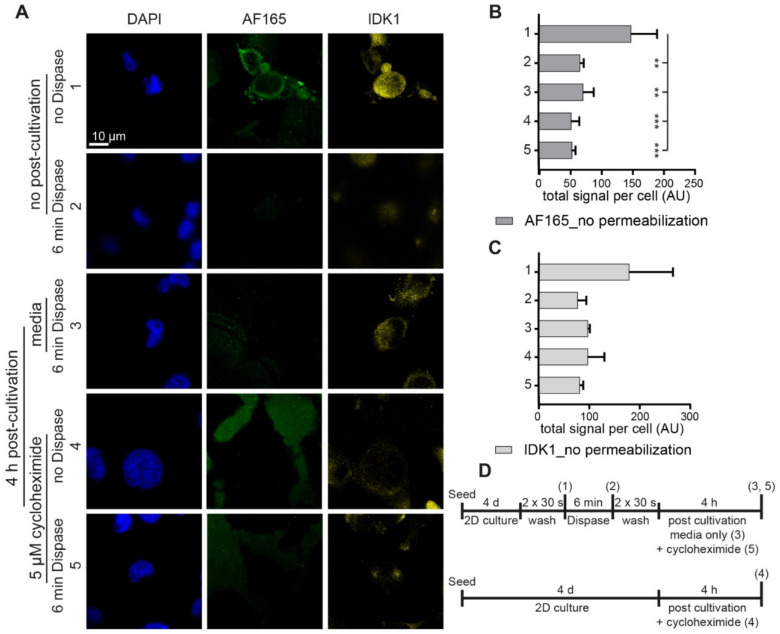
In the absence of permeabilization, BSP immunofluorescence levels are diminished upon dispase exposure. 1,200,000 MDA-MB-231 cells were seeded in Petri dishes containing glass cover slips. After 4 d, cultures were treated with dispase and partially further cultivated with media or for secondary treatments with cycloheximide after washing, as indicated. PFA fixed and non-permeabilized samples were stained for DAPI and human BSP (AF165 and IDK1). (**A**) Representative confocal images in sum projections. (**B**,**C**) Graphs showing quantitative analysis of fluorescence intensity per cell. See (**D**) for methodology. Mean + SD (*n* = 3; ** *p* < 0.01, *** *p* < 0.001).

**Figure 4 cells-10-01304-f004:**
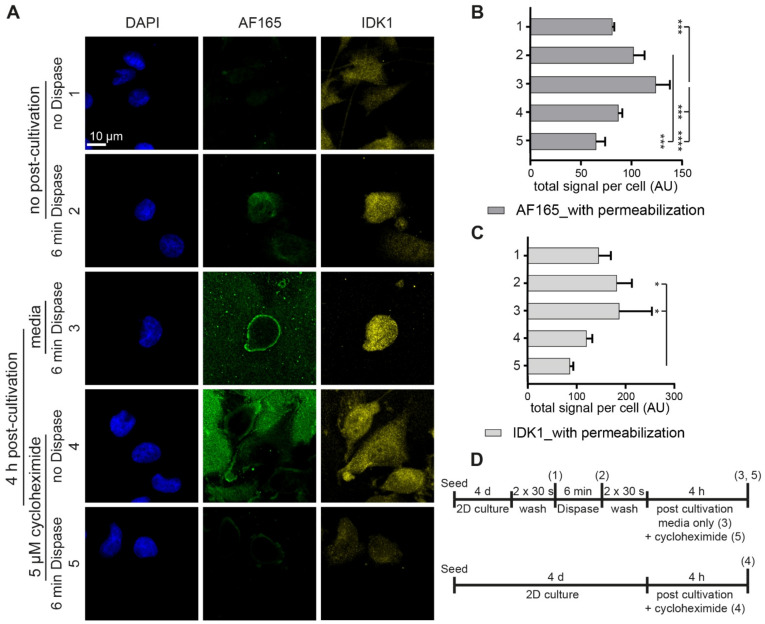
Dispase induces BSP biosynthesis in MDA-MB-231 cells. 1,200,000 MDA-MB-231 cells were seeded in Petri dishes containing glass cover slips. After 4 d, cultures were treated with dispase and partially further cultivated with media or for secondary treatments with cycloheximide after washing as indicated. PFA fixed samples were permeabilized and then stained for DAPI and BSP (AF165 and IDK1). (**A**) Representative confocal images in sum projections. (**B**,**C**) Graphs showing quantitative analysis of fluorescence intensity per cell. See (**D**) for methodology. Mean + SD (*n* = 3; * *p* < 0.05, *** *p* < 0.001, **** *p* < 0.0001).

**Figure 5 cells-10-01304-f005:**
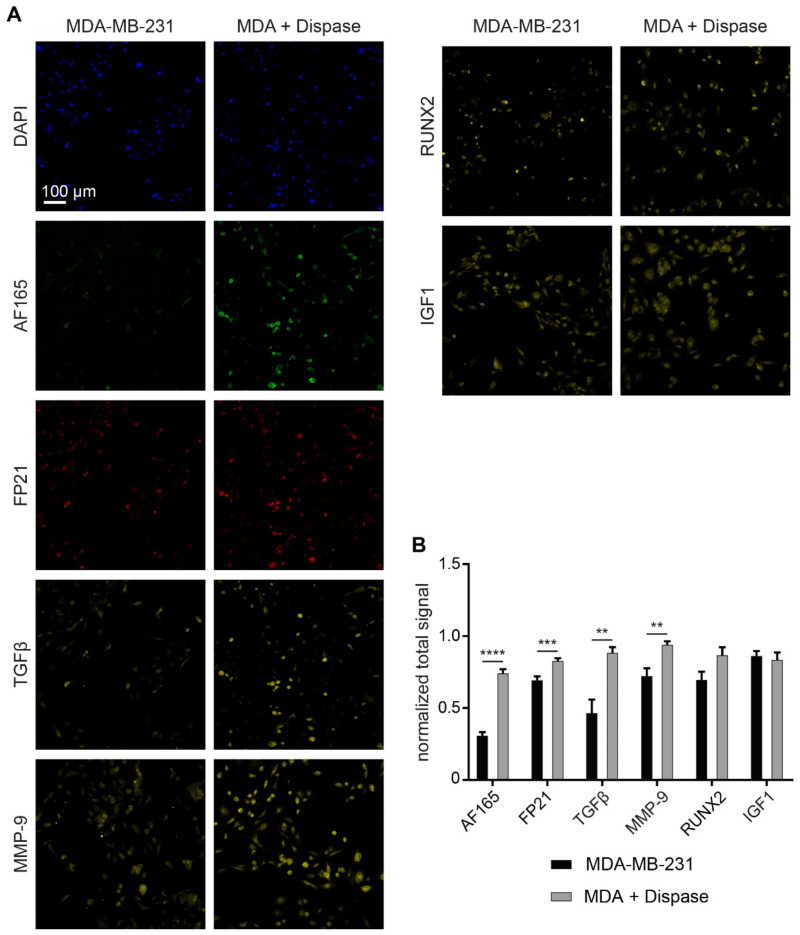
Immunofluorescence signals of regulatory markers are consistent with induced BSP expression. 1,200,000 MDA-MB-231 cells were seeded in Petri dishes containing glass cover slips. After 4 d, cultures were treated with dispase for 6 min. PFA fixed samples were permeabilized and then stained for DAPI, BSP (AF165 and FP21), and one additional regulatory marker (TGFβ, MMP-9, RUNX2 or IGF1). (**A**) Representative confocal images in sum projections. (**B**) Graph showing quantitative analysis of fluorescence intensity per cell. Mean + SEM (*n* = 6 for TGBβ, MMP-9, RUNX2, IGF1; *n* = 24 for AF165, FP21; ** *p* < 0.01, *** *p* < 0.001, **** *p* < 0.0001).

**Figure 6 cells-10-01304-f006:**
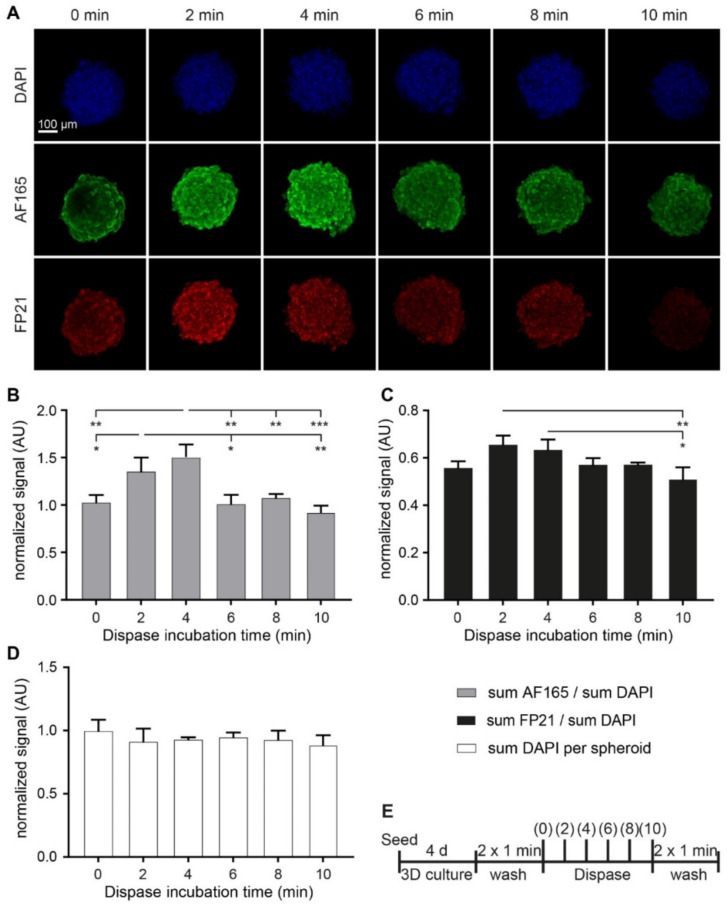
Dispase leads to a transient increase of anti-BSP immunofluorescence in MDA-MB-231 spheroids. 4000 MDA-MB-231 cells were seeded in mono-culture spheroids and supplemented with 2.5% BME upon seeding. After 4 d, spheroids were exposed to dispase for different incubation times of up to 10 min as indicated. PFA fixed spheroids were stained for DAPI and human BSP (AF165 and FP21). (**A**) Representative confocal images in volume projections. (**B**–**D**) Graphs show quantitative analysis of sum immunofluorescence intensities over the entire spheroid normalized to sum fluorescence intensities of DAPI (**B**,**C**) or sum of DAPI fluorescence signal per spheroid (**D**). (**E**) Methodological overview. Mean + SD (*n* = 3; * *p* < 0.05, ** *p* < 0.01, *** *p* < 0.001).

**Figure 7 cells-10-01304-f007:**
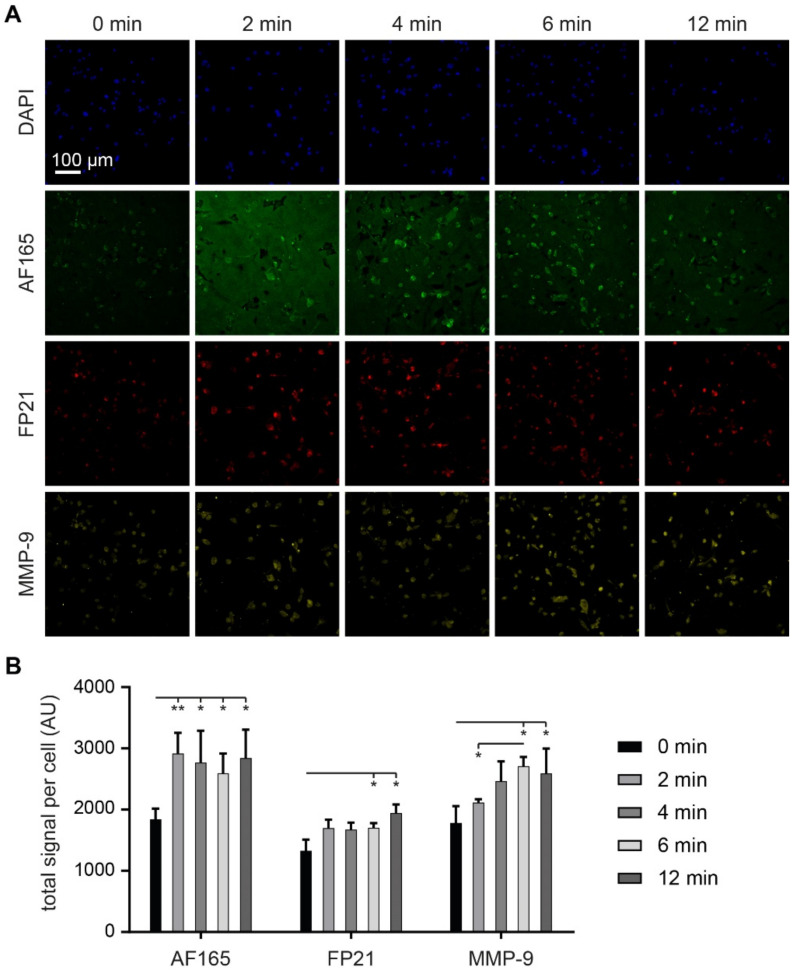
Incubation with MMP-9 protein enhances anti-BSP immunofluorescence. 1,200,000 MDA-MB-231 cells were seeded in Petri dishes containing glass cover slips. After 4 d, cultures were treated with 400 ng/mL MMP-9 in FCS-free media for exposure times as indicated, or with fresh media as 0 min control. Then, samples were fixed and stained for DAPI and human BSP (AF165 and FP21) as well as MMP-9. (**A**) Representative confocal images in sum projections. (**B**) Graphs showing quantitative analysis of fluorescence intensities per cell. Mean + SD (*n* = 3; * *p* < 0.05, ** *p* < 0.01).

## Data Availability

All results generated or analyzed during the present study are included in this published article. Data and materials will be made available upon request via email to the corresponding author (r.rudolf@hs-mannheim).

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
