# Peer review of "Extracellular Matrix Components Regulate Bone Sialoprotein Expression in MDA-MB-231 Breast Cancer Cells"

_cells, 2021, doi:10.3390/cells10061304_

Round 1

Reviewer 1 Report

The manuscript "Extracellular matrix components regulate bone
sialoprotein expression in MDA-MB-231 breast cancer cells"  describes the effects of ECM components like BME and collagen I on the expression of BSP in a breast cancer cell line. Although this is an interesting topic, I have some major concerns regarding the results and interpretation, which should be addressed. The methods employing 2D and 3D spheroid culture is very innovativ, however some other apporaches as well as the presentation and interpretation of results raise questions.
I agree with the authors that basement membrane extract induces the expression of distinct types of BSP protein.

  1. The figures 2-5 show fluorescence stainings but are kept in black and white, which is in my opinion the wrong format. In these images the described effects are hardly visible, especially since there at least sometimes are only one or two cells visible and no differences can be seen. An additional overview image might give a better impression. But the main criticism is the black and white image that should be changed to a coulered image. Another question is how many cells were included in the quantification- this is not described anywhere.
  2. In this work, 3 different antibodies were used - why are always 2 different antibodies used in different experiments? That should at least be described in the discussion and also how these antibodies differ from each other.
  3. Figure 5: Why is the sum AF165/DAPI above 1 and the sum FP21/DAPI below? What does this show? This should be discussed.
  4. Why was dispase used? I would have recommended to use a protease that is found in the ECM, e.g. MMPs. 
  5. My main concern is the interpretation of dispase activity: Does proteolytic affinity directly influence protein expression? Proteases degrade ECM components and activate other proteases that further degrade proteins. As far as I know no direct influence on protein de novo synthesis has been described. The authors state that dispase enhances BSP de novo synthesis - after incubation with dispase in 6 minutes. This is not possible - protein de novo synthesis takes much longer than six minutes. The authors state that this is proven by cycloheximid inhibition - but cycloheximid inhibits protein synthesis anyway and is not a competitor to dispase.
  6. An explanation for the higher BSP staining after dispase incubation might be that dispase hydrolyzes peptide bonds and more fragments of BSP result and maybe thereby  access of antibodies is facilitated- this might also explain the different results after staining with different antibodies.
  7. The discussion is very short and very superficial. Hardly any results are discussed in comparison to existing literature.

Overall the working hypothesis, the results and discussion have to be revised. The interpretaion of the results of the dispase incubation is in my opinion wrong and should be reconsidered.

Author Response

We want to thank the reviewer for the critical and constructive analysis of our article. In the following, please find the point-to-point responses to the specific concerns:

  1. The figures 2-5 show fluorescence stainings but are kept in black and white, which is in my opinion the wrong format. In these images the described effects are hardly visible, especially since there at least sometimes are only one or two cells visible and no differences can be seen. An additional overview image might give a better impression. But the main criticism is the black and white image that should be changed to a coulered image. Another question is how many cells were included in the quantification- this is not described anywhere.

>>Figures have now been converted to color coded ones. For the MDA-MB-231 alone and Dispase treated conditions, further 20x overviews were added in fig. 5 and 7. The number of quantified experiments is shown in the corresponding figure legends, the number of analyzed cells has now been added in the methods (at least 10 cells per experiment, see lines 162-163).

  1. In this work, 3 different antibodies were used - why are always 2 different antibodies used in different experiments? That should at least be described in the discussion and also how these antibodies differ from each other.

>> The three employed antibodies are from different species: AF165 – human, FP21 – mouse, IDK1 – rat. Primarily, different anti-BSP antibodies were used to consolidate the results. In addition, given that AF165 and IDK1 showed rather cell surface/extracellular localization and FP21 presumably in the Golgi apparatus, FP21 was not useful for experiments without cell permeabilization. This has been described (see lines 226-231 & 338-351). To illustrate this, the new Fig. S1 (see lines 559-566) and S2 (see lines 568-580) have been added.

  1. Figure 5: Why is the sum AF165/DAPI above 1 and the sum FP21/DAPI below? What does this show? This should be discussed.

>> Maybe, the description here was not precise enough. The depicted data show mean values of the sum immunofluorescence signals per spheroid divided by the sum fluorescence intensity per spheroid of DAPI. This kind of quantification was chosen to normalize the immunofluorescence signals to the cell number. Such kind of analysis would indicate relative differences of immunofluorescence intensity. In the present case, this would mean that AF165 immunofluorescence signals are, on average, stronger than those of FP21. But this has no biological relevance, since it may mainly reflect intrinsic antibody signal strength and imaging settings. Thus, the only relevant information comes from comparing effects of different experimental conditions on immunofluorescence signals of one given antibody. And this showed a rise of AF165 and FP21 signals upon Dispase treatment of up to four minutes. The corresponding figure legend (now fig. 6) was rephrased (see lines 296-298). Hopefully, this is now clearer.

  1. Why was dispase used? I would have recommended to use a protease that is found in the ECM, e.g. MMPs.

>> We perfectly agree that MMP-induced signals would enable a more straightforward interpretation. Therefore, we added confirming experiments with MMP-9 treatment (fig. 7, see lines 300-316). Incubating MDA-MB-231 cultures with 400 ng/mL active MMP-9 protein for 2-12 min resulted in a significant increase of anti-BSP fluorescence signals compared to treatment control cultures. Anti-MMP-9 staining quantification correlated with protein incubation times.

  1. My main concern is the interpretation of dispase activity: Does proteolytic affinity directly influence protein expression? Proteases degrade ECM components and activate other proteases that further degrade proteins. As far as I know no direct influence on protein de novo synthesis has been described. The authors state that dispase enhances BSP de novo synthesis - after incubation with dispase in 6 minutes. This is not possible - protein de novo synthesis takes much longer than six minutes. The authors state that this is proven by cycloheximid inhibition - but cycloheximid inhibits protein synthesis anyway and is not a competitor to dispase.

>> We thank the reviewer for this comment, because it allowed us to cover these concerns in the discussion. Indeed, it is known from literature that protease-activated receptors (PAR) may be activated by trypsin, proteinase 3, or MMP-9. These may activate ERK and Ras signaling via G-Protein signal transduction. Since MDA-MB-231 cells do express PAR-2 receptor and its’ signaling was shown to induce cellular reactions within 2 min and protein synthesis within 5 min, we think that the timelines of our experiments are compatible with the induction of acute protein expression. The discussion section was expanded accordingly (see lines 364-370, 375-390). Furthermore, it might be important to clarify the timeline between the start of Dispase exposure and PFA fixation. Indeed, since the enzymatic activity had to be stopped at the end of Dispase treatment, the cultures were first washed with FCS-containing media after Dispase aspiration. Then, the samples were washed two more times with PBS for 30 s (2D) or 1 min (3D) each. In summary, there were roughly two to three more minutes for the cells to express induced proteins before terminal culture fixation. To make this clear, the corresponding descriptions were expanded in the methods, results, and discussion sections (see lines 103f., 106f., 109f., 207f., 370f.). As for cycloheximide, we do not claim an inhibition based on competitive receptor binding, inhibition of Dispase, or comparable effects, but we think that the blocking of protein synthesis per se does also include de novo synthesis of BSP protein. Otherwise, this effect is very unlikely to be prohibited by the presence of cycloheximide. We also added a short remark on this in the discussion (see lines 361f.).

  1. An explanation for the higher BSP staining after dispase incubation might be that dispase hydrolyzes peptide bonds and more fragments of BSP result and maybe thereby access of antibodies is facilitated- this might also explain the different results after staining with different antibodies.

This certainly a valid point. In fact, we cannot exclude that Dispase treatment enhanced the accessibility of anti-BSP antibody AF165 to extracellular BSP. Therefore, this effect might support AF165-based signal intensity increase. But since increasing signals were also found for FP21 staining and since this is intracellular, we think that our explanations on this point are supported by experimental evidence. A corresponding section has been added to the discussion to highlight these points (see lines 372-374).

  1. The discussion is very short and very superficial. Hardly any results are discussed in comparison to existing literature.

Following the referee’s concerns, we expanded the discussion part and included additional literature for comparison and validation (see lines 365-390).

Reviewer 2 Report

Title: Extracellular matrix components regulate bone sialoprotein expression in MDA-MB-231 breast cancer cells.

Your article is rather novel. The effects of extracellular matrix components on BSP expression were detected by 2D and 3D cells. And we have some opinions on the content of the article:

  • The content of the article is a little monotonous. In addition to detecting the expression of BSP by fluorescence, whether there are other detection indicators to be demonstrated.
  • Have you considered detecting the effect of up-regulation of BSP on other pathway molecules? Such as ERK, P38, etc.
  • The 3D cell model is more consistent with the survival pattern of cancer cells. But we have a few questions about the experiment with the 3D cell model. May I ask whether the 3D cell ball should be carried out with the scaffold or stripped off the cultured 3D cell ball for the experiment? If you do it with the scaffold, how do you make sure that all the cells in the scaffold are treated, if you peel the SD cell balls off the scaffold and do the experiment, how do you make sure that the balls are clean and intact.

Author Response

We thank the reviewer for the positive feedback. Please, see the following point-to-point response to specific comments.

  1. The content of the article is a little monotonous. In addition to detecting the expression of BSP by fluorescence, whether there are other detection indicators to be demonstrated.

>> The immunofluorescence-based analysis was chosen since we tried to include subcellular information. Thus, instead of adding other detection methods like SDS-PAGE, we tried to validate our findings by analyzing different culture conditions. In our opinion, parallelized immunofluorescence staining, as used in this study, is robust and should comply with the aims of the work.

  1. Have you considered detecting the effect of up-regulation of BSP on other pathway molecules? Such as ERK, P38, etc.

We agree that including additional regulating proteins would strengthen the manuscript and thank the reviewer for the opportunity to make this point. Therefore, we added fig. 5 and corresponding passages in the results section showing the analysis of RUNX2, IGF1, TGFβ, and MMP-9 in the presence and absence of Dispase treatment (see lines 260-274). While TGFβ and MMP-9 showed significant increases upon Dispase exposure, RUNX2 revealed a trend to increase and IGF1 did not change.

  1. The 3D cell model is more consistent with the survival pattern of cancer cells. But we have a few questions about the experiment with the 3D cell model. May I ask whether the 3D cell ball should be carried out with the scaffold or stripped off the cultured 3D cell ball for the experiment? If you do it with the scaffold, how do you make sure that all the cells in the scaffold are treated, if you peel the SD cell balls off the scaffold and do the experiment, how do you make sure that the balls are clean and intact.

For the 3D experiments, no pre-assembled scaffold structure was applied. Rather, the BME was mixed with the cells upon seeding to help with intercellular adherence and spheroid assembly. Due to the homogeneous distribution within the intercellular space in the spheroid, a specific removal during the harvesting was not applicable. We validated the applicability of our analysis through spheroid slices, revealing that while AF165 signal is rather located on the spheroid rim, FP21-signal is more evenly distributed on many cells of the spheroid (fig. S2; see lines 568-579). Therefore, we think that the similar trends in fig. 6 for both anti-BSP antibody immunofluorescence levels are in support of a Dispase-induced upregulation of BSP signals.

Round 2

Reviewer 1 Report

The manuscript improved substantially and can be accepted in its present form.

Reviewer 2 Report

The revised manuscript can be accepted for publication.